# Bounded Rationality for LLMs: Satisficing Alignment at Inference-Time

**Mohamad Chehade** [* 1]   **Soumya Suvra Ghosal** [* 2]   **Souradip Chakraborty** [2]   **Avinash Reddy** [3]   **Dinesh Manocha** [2]
**Hao Zhu** [1]   **Amrit Singh Bedi** [3]

## Abstract

Aligning large language models with humans is challenging due to the inherently multifaceted nature of preference feedback. While existing approaches typically frame this as a multi-objective optimization problem, they often overlook how humans actually make decisions. Research on bounded rationality suggests that human decision-making follows satisficing strategies—optimizing primary objectives while ensuring others meet acceptable thresholds (Simon, 1956). To bridge this gap and operationalize the notion of satisficing alignment, we propose `SITAlign`: an inference-time framework that addresses the multifaceted nature of alignment by maximizing a primary objective while satisfying threshold-based constraints on secondary criteria. We provide theoretical insights by deriving suboptimality bounds of our satisficing-based inference alignment approach. We empirically validate `SITAlign`'s performance through extensive experimentation on multiple benchmarks. For instance, on the PKU-SafeRLHF dataset with the primary objective of maximizing helpfulness while ensuring a threshold on harmlessness, `SITAlign` outperforms the state-of-the-art multi-objective decoding strategy by a margin of 22.3% in terms of GPT-4 win-tie rate for helpfulness reward while adhering to the threshold on harmlessness.

## 1. Introduction

Aligning large language models (LLMs) with human preferences is crucial for their safe, helpful, and effective deployment. However, these preferences are inherently multifaceted, requiring consideration of multiple attributes like safety, helpfulness, truthfulness, and conciseness, often captured by multiple reward models (Bai et al., 2022b; Dai et al.; Maas et al., 2011). Prior alignment literature (Jang et al., 2023; Shi et al., 2024) has approached the problem through the lens of multi-objective optimization, ignoring the underlying special structure characterizing human decision making or preferences. Additionally, this traditional formulation faces significant challenges: determining appropriate weights for different, often conflicting, objectives is difficult, and more fundamentally, it assumes that all preference dimensions should be simultaneously maximized.

**Beyond traditional alignment.** Drawing inspiration from theories of human decision-making, particularly the satisficing principles from bounded rationality (Simon, 1956)—we propose an alternate perspective for multi-faceted LLM alignment. Satisficing theory suggests that humans often do not seek to maximize every objective; instead, they adopt strategies that prioritize optimizing key goals while ensuring other important objectives simply meet acceptable thresholds. For example, one might prioritize finding a good enough solution quickly rather than searching exhaustively for the absolute best. We argue that this principle translates naturally to LLM alignment: it is often sufficient to maximize a primary objective, such as helpfulness or relevance, while ensuring other attributes, like harmlessness, bias, or verbosity, stay within acceptable thresholds (cf. Section 3.1).

**Satisficing Alignment.** The 'satisficing alignment' paradigm represents a significant departure from traditional multi-objective maximization approaches. While the latter aims for optimal trade-offs across all objective dimensions simultaneously, often via a single scalar objective obtains by weighted combination (Shi et al., 2024; Son et al., 2025), a satisficing approach explicitly acknowledges that some objectives behave as constraints to be met rather than targets to be continuously improved. The existing alignment research has largely overlooked this satisficing perspective, focusing predominantly on methods that combine or maximize multiple reward signals without explicitly incorporating threshold-based constraints derived from human decision-making principles. Implementing such a flexible, threshold-based alignment strategy through traditional finetuning is challenging, especially since acceptable thresholds can vary

---

[*]Equal contribution   [1]University of Texas at Austin, Austin, TX, USA   [2]University of Maryland, College Park, MD, USA   [3]University of Central Florida, Orlando, FL, USA. Correspondence to: Amrit Singh Bedi <amritbedi@ucf.edu>.

*Proceedings of the 42$^{nd}$ International Conference on Machine Learning*, Vancouver, Canada. PMLR 267, 2025. Copyright 2025 by the author(s).

widely across users, contexts, and tasks. This necessitates an adaptive, fully inference-time approach. To address this, we introduce SITAlign —a satisficing alignment framework that maximizes a primary reward while enforcing thresholds on secondary objectives at inference time. Our method avoids costly model finetuning by enabling a fully inference-time approach that offers a dynamic and personalized way to control LLM outputs in accordance with the satisficing principle. We summarize our key contributions as follows.

1. **Satisficing alignment.** We introduce the concept of satisficing alignment as an alternate perspective for handling multiple criteria in LLM control, distinct from traditional multi-objective maximization. We characterize the alignment problem of handling multi-faceted user preferences by drawing connections to bounded rationality and human satisficing strategies, where individuals maximize key objectives while ensuring that others meet acceptable thresholds.

2. **We propose SITAlign**, an inference-time framework that operationalizes the content of satisficing alignment by allowing maximization of a primary objective under constraints derived from thresholds on secondary objectives. Our proposed approach is based on utilizing ideas from duality theory (Nedić & Ozdaglar, 2009).

3. **Theoretical insights of SITAlign.** We theoretically analyze the suboptimality of our proposed method and derive performance bounds in terms of primal as well as dual variables.

4. **Experimental Evaluations:** For empirical validation, we compared our approach with various baseline (Khanov et al., 2024) and state-of-the-art multi-objective decoding strategies (Shi et al., 2024) across three evaluation setups. Our analysis in Section 6 reveals that SITAlign outperforms competing approaches in terms of the GPT-4 win-tie rate across all setups. For example, when optimizing the primary objective of helpfulness while adhering to the humor threshold in generated responses, SITAlign improves the win-tie rate by approximately $10\%$ compared to the current state-of-the-art (Shi et al., 2024).

## 2. Related Works

**Alignment in LLMs.** The common approach in LLM alignment is reinforcement learning from human feedback (RLHF), where a reward model is first learned from human preferences, and the proximal policy optimization (PPO) algorithm is then used to derive the aligned policy (Bai et al., 2022a; Askell et al., 2021; Glaese et al., 2022; Ouyang et al., 2022). Although widely used, PPO has been reported to suffer from instability and high computational

demand. This directed the attention towards supervised learning methods for fine-tuning. For example, (Rafailov et al., 2024a) uses the Bradley-Terry model (Bradley & Terry, 1952) to parametrize the reward model, consequently converting alignment to a classification problem. Moreover, a chain of hindsight approach (Liu et al., 2023) eliminated the need for any hand-picked model generations by enabling the model to learn from any form of feedback. (Faiz et al., 2023) use a ranking loss to align the model probabilities of responses while (Dong et al., 2023) suggest supervised fine-tuning on the highest reward samples. The self-play tuning of (Chen et al., 2024) even removes the necessity for any human-annotated dataset. Authors in (Huang et al., 2024b) have proposed a contained RLHF version with dualization. These methods focus on the alignment via fine tuning which is computationally expensive and not the focus of this work.

**Inference-time Alignment of LLMs.** A simple and effective inference-time method is known as the best-of-$K$ (Stiennon et al., 2020a; Nakano et al., 2021; Touvron et al., 2023), where $K$ iid samples are drawn from a base model, and the sample with the highest reward is generated. Controlled decoding methods, on the other hand, generate responses one token at a time. Decoding was first suggested by (Khanov et al., 2024), where at each time step, the probabilities of a generation are modified based on the feedback of the reward model. In addition, (Huang et al., 2024a) model text generation as a search process, with a state space made up of the sequence of tokens, and an action space of the vocabulary of words. The most notable decoding work, nevertheless, appears in (Mudgal et al., 2023) and approximates decoding by collecting samples from a reference model. (Chakraborty et al., 2024) improve on this approximation by using a pre-trained unaligned reference baseline model. While such methods can solve the alignment problem efficiently, they can only meet one user preference at a time, and therefore, the user can only provide a single criterion for the alignment. (Shi et al., 2024) solve this problem with their multi-objective formulation for decoding. This formulation, nonetheless, treats all of the criteria as part of a weighted objective function, rather than minimum requirements that must be met. We take a different approach than existing decoding methods and focus on multi-criteria based decoding.

## 3. Problem Formulation

### 3.1. Our Key Insight: Satisficing Alignment

To understand the potential of satisficing alignment, we begin with a key insight: beyond a certain point, maximizing reward scores might not be necessary to satisfy human preferences. Instead, we hypothesize that there exists a threshold for certain attributes (e.g., harmlessness) where responses scoring above that threshold are often deemed

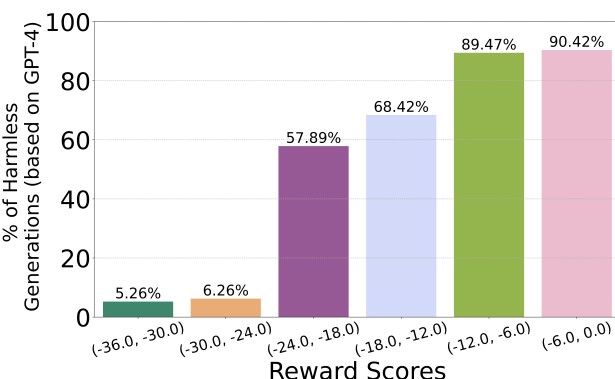

*Figure 1.* This figure shows the percentage of responses from LLM being harmless if there reward score lies in particulate range shown on the x-axis. We use GPT-4 evaluations to decide if the response is harmless of not. This clearly shows that approximately 90% of the responses are harmless if reward score is more than $-12$.

acceptable, effectively satisfying the user's requirements. Hence, instead of striving for the absolute "best" answer on every dimension, our core concept revolves around the idea that ensuring thresholds are met for secondary objectives, is often sufficient to deliver human-aligned outputs.

**Proof of Concept**: To validate this intuition, and demonstrate the practicality of our approach, we conducted a proof-of-concept experiment. We aimed to illustrate that, in practice, meeting a specific reward threshold can lead to outputs that satisfy human preferences. Specifically, we focused on the attribute of harmlessness. We consider prompts from the test set of PKU-SafeRLHF dataset (Ji et al., 2024) and generate $N = 20$ responses $\{\mathbf{y}\}_{i=1}^{N}$ using a Zephyr-7B-$\beta$[1] model. We evaluate the harmlessness of each response using a pre-trained harmless reward model[2]. We divide the reward scores, which range between $[-36, 0]$, into six equal bins (cf. Figure 1) and ensure that each bin contains $N$ responses. For each bin, we assess the percentage of harmless responses via GPT-4. As shown in Figure 1, the results confirm that as the reward scores increase, the proportion of harmless responses also rises. Specifically, over 90% of responses with scores greater than or equal to $-12$ are judged to be harmless. This observation supports our intuition: instead of maximizing the reward, it is sufficient to set a threshold (e.g., $-12$) and ensure that generated responses exceed this value. This motivates our satisficing inference time alignment framework, which we formulate next.

**Selection of threshold.** A natural question arises: how should we choose the threshold? Interestingly, determining an appropriate threshold is not particularly challenging in practice for a given reward model. One effective approach is

to leverage GPT-4 win rates to estimate a reasonable threshold a priori. Additionally, if human feedback is available, it can serve as a valuable resource for refining the threshold selection.

### 3.2. Satisficing Inference-Time Alignment

We formulate this alignment problem via utilizing the controlled decoding procedure (Khanov et al., 2024; Mudgal et al., 2023) which we describe here in detail. For a given prompt $\mathbf{x}$, an LLM generates a response $\mathbf{y} = [y_0, y_1, ..., \text{EOS}]$ by sampling $y_t \sim \pi_{\text{sft}}(\cdot|[\mathbf{x}, \mathbf{y}_{<\mathbf{t}}])$ at any time step $t$. This token-by-token generation of the response is known as *decoding*. The problem can be modeled as Markov decision process (MDP) (Puterman, 2014), with tuple $(\mathcal{S}, \mathcal{A}, \mathcal{P}, r, \gamma)$. Given a time step $t$ in the decoding process of LLM, the state $s_t \in \mathcal{S}$ is the concatenation $[\mathbf{x}, \mathbf{y}_{<t}]$, where $\mathbf{x}$ is the initial prompt provided by the user, and $\mathbf{y}_{<\mathbf{t}}$ is the response generated by LLM up until $t$. The LLM then decide the next token $y_t \in \mathcal{A}$ $y_t \sim \pi(\cdot|s_t)$. Once $y_t$ is determined, it is concatenated to $s_t$ to form $s_{t+1} = [\mathbf{x}, \mathbf{y}_{<\mathbf{t}}, y_t]$. Therefore, all transitions $\mathcal{P}$ are deterministic.

**Reward function.** We generate a response $\mathbf{y}$ from decoding policy $\pi$. The goodness of such a response is quantified by the reward function $r(\mathbf{x}, \mathbf{y})$ which evaluates given prompt $\mathbf{x}$ and full trajectory response $\mathbf{y}$. Since our policy $\pi$ is token level, we can write a corresponding *trajectory-level policy* as $\rho_\pi(\mathbf{y}|\mathbf{x}) = \prod_{t=1}^{T} \pi(y_t|\mathbf{x}, \mathbf{y}_{<t})$.

**The action-value function:** At each step $t$, the value of the given state $s_t$ (which is the response generated so far) and current action token $a_t$ can be measured by the expected value of the reward to be received at the end of the sequence, denoted by the *action-value function* $Q^\pi$:

$$Q^\pi(s_t, z)z = \mathbb{E}_{\tau \sim \rho_\pi(\cdot|s_t, z)}\big[r([\mathbf{x}, \mathbf{y}_{<t}, z], \tau)\big], \quad (1)$$

where $\tau$ denotes the trajectory $\tau := [z_1, z_2, \cdots, z_T]$ sampled from $\rho_\pi(\cdot|s_t, z)$. Hence, we can write the optimal Q-function is given by $Q^*(s_t, z_t) = \max_\pi Q^\pi(s_t, z_t)$.

**Objective.** Now we are ready to present the optimization problem of this work. We formulate it as a constrained controlled decoding problem:

$$\pi_{\text{dec}}^*(\cdot|s_t)$$
$$:= \arg\max_{\pi \in \Pi} \mathbb{E}_{a \sim \pi(\cdot|s_t)}\Big[Q_1^*(s_t, z)\Big] \quad (2)$$
$$- \beta_1 \mathcal{D}_{\text{KL}}\Big[\pi(\cdot|s_t) \,\|\, \pi_{\text{sft}}(\cdot|s_t)\Big],$$
$$\text{subject to } \mathbb{E}_{z \sim \pi(\cdot|s_t)}\Big[Q_2^*(s_t, z)\Big] \geq \beta_2,$$
$$\vdots$$
$$\mathbb{E}_{z \sim \pi(\cdot|s_t)}\Big[Q_N^*(s_t, z)\Big] \geq \beta_N.$$

---

[1]HuggingFaceH4/zephyr-7b-beta
[2]Skywork/Skywork-Reward-Llama-3.1-8B-v0.2

where $Q_i^*(s_t, z) := \mathbb{E}_{\tau \sim \rho^*}\left[r_i([s_t, z], \tau)\right]$ are the action-value functions of the reward functions $i = 1, \dots i = N$, and $[\beta_2, \dots \beta_N]$ are pre-defined thresholds. We note that the problem in (2) is a generalization of controlled decoding formulation in Mudgal et al. (2023); Chakraborty et al. (2024), which is for unconstrained scenarios.

# 4. Proposed Approach

In this section, we propose a method for solving the constrained optimization problem in (2) via duality theory. Thanks to the strongly convex objective and linear constraints in $\pi$, the overall problem is strongly convex. A convenient step is then to write the Lagrangian function of the problem:

$$\mathcal{L}([x, y^t]; \pi, \lambda) = \sum_{i=1}^{N} \lambda_i \mathbb{E}_{z \sim \pi(\cdot|[x,y^t])}\left[Q_i^\pi([x, y^t], z)\right]$$
$$- \beta_1 D_{\text{KL}}(\pi \| \pi_0) - \sum_{i=2}^{N} \lambda_i \beta_i, \quad (3)$$

where $\lambda \in \mathbb{R}_+^N$ is the vector of Lagrange multipliers, with $\lambda_1 = 1$. $\pi$ and $\lambda$ are the primal and dual variables of the optimization problem, respectively. While the former is the optimal decoding policy at a given state $[x, y^t]$, the latter represents the sensitivity of the objective to the changes in the thresholds $\beta_i$. Particularly, $\lambda_i = 0$ means the corresponding constraint (with $\beta_i$ threshold) is satisfied with strict inequality. The optimal primal-dual pair $(\pi^*, \lambda^*)$ is the solution to the following optimization problem:

$$\max_{\pi} \min_{\lambda \in \mathbb{R}_+^N} \mathcal{L}([x, y^t]; \pi, \lambda) = \min_{\lambda \in \mathbb{R}_+^N} \underbrace{\max_{\pi} \mathcal{L}([x, y^t]; \pi, \lambda)}_{\mathcal{L}([x,y^t]; \pi^{*,\lambda}, \lambda)}. \quad (4)$$

**Primal variable.** Therefore, for a given $\lambda \in \mathbb{R}_+^N$, the optimal primal variable $\pi^{*,\lambda} := \arg\max_\pi \mathcal{L}([x, y^t]; \pi, \lambda)$ is given by:

$$\pi^{*,\lambda}(z|[x, y^t]) \quad (5)$$
$$:= \frac{\pi_{\text{sft}}(z|[x, y^t])}{Z_\lambda([x, y^t])} \exp\left[\frac{1}{\beta_1} \sum_{i=1}^{N} \lambda_i Q_i^{\pi^{*,\lambda}}([x, y^t], z)\right],$$

where $Z_\lambda$ is a normalizing factor. The derivation can be found in Appendix B.

**Dual problem.** Eq. (5) indicates that for every $\lambda$, a new $\pi^{*,\lambda}$ exists. However, there exists a unique optimal primal variable $\pi^*$, which corresponds to $\lambda^*$, the optimal dual variable, i.e. $\pi^* := \pi^{*,\lambda^*}$. Having expressed $\pi$ in terms of $\lambda$, we can find $\lambda^*$ by solving the following optimization problem in $\lambda$:

$$\lambda^* := \arg\min_{\lambda \in \mathbb{R}_+^N} \mathcal{L}([x, y^t]; \pi^{*,\lambda}, \lambda)$$
$$= \beta_1 \log\left(\mathbb{E}_{z \sim \pi_{\text{sft}}(\cdot|[x,y^t])}\left[\exp\left(\frac{1}{\beta_1} \sum_{i=1}^{N} \xi_i(\lambda)\right)\right]\right)$$
$$- \sum_{i=2}^{N} \lambda_i \beta_i, \quad (6)$$

where $\xi_i(\lambda) = \lambda_i Q_i^{\pi^{*,\lambda}}([x, y^t], z)$.

**Challenges.** Although the optimal primal and dual variables are given by (5) and (6), finding $\pi^*$ and $\lambda^*$ is challenging for two reasons:

**(1) Computing $\lambda^*$ is expensive:** Despite the strong convexity, the computational resources at inference time do not allow for solving the problem using an iterative algorithm like projected gradient descent (Huang et al., 2024b). Instead, a closed-form solution is needed.

**(2) $Q^*$ is not available:** the analysis above assumes knowledge of the optimal action-value function $Q^{\pi^*,\lambda}$, which is very difficult to compute (Mudgal et al., 2023), especially during the optimization process.

To tackle these challenges, we propose the following methods to estimate $\lambda^*$ and $Q^*$:

**(1) Estimating $\lambda^*$.** A closed-form solution for (6) is difficult to find, but becomes possible when we consider a quadratic approximation of the objective function. The solution is then given by:

$$\lambda^* = \left[\left(\left[\nabla_\lambda^2 Z_\lambda([x, y^t])\right]_{\lambda=0}\right)^{-1} \right.$$
$$\left. \times \left(\beta - \left[\nabla_\lambda Z_\lambda([x, y^t])\right]_{\lambda=0}\right)\right]^+. \quad (7)$$

where $\beta \in \mathbb{R}^N$ and $[\cdot]^+$ denotes projection onto the positive orthant. A more detailed derivation can be found in Appendix B.

**(2) Estimating $Q^*$.** Recent advancements have developed very accurate estimates of $Q^*$, the most notable of which is Transfer $Q^*$, or $\text{TQ}^*$ (Chakraborty et al., 2024), which we follow in our work. In short, $\text{TQ}^*$ (given in (8)) relies on sampling trajectories from trajectory-level baseline policy $\rho_i^{\text{BL}}$, which has been pre-trained yet not aligned with the new criteria.

$$\text{TQ}_i^\star(s_t, z) = \mathbb{E}_{\tau \sim \rho_i^{\text{BL}}(\cdot|s_t, z)}\left[r_i([s_t, z], \tau)\right] \quad (8)$$

Finally, our approach, denoted by $\text{SITAlign}$, or Satisficing Inference-Time Alignment of Large Language Models, is summarized in Algorithm 1.

**Algorithm 1** `SITAlign`: Satisficing Inference-Time Alignment of Large Language Models

---

1: **Input:** Trajectory level baseline model $\rho_i^{\text{BL}}(\mathbf{y}|\mathbf{x})$ aligned with baseline reward $r_i^{\text{BL}}$, set of rewards $r_i$, $i = \{1, ..., N\}$, where $r_1$ is the target reward, a vector of parameters $\beta$, token-level baseline policy $\pi_{\text{BL}}$, number of tokens sampled $k$, decoding alignment parameter $\alpha$, vocabulary set $\mathcal{V}$.
2: **for** $t = 0, \ldots, T$ **do**
3:   Current state: $s_t = [\mathbf{x}, \mathbf{y}_{<t}]$, where $\mathbf{x}$ is the prompt and $\mathbf{y}_{<t} = [y_0, y_1, \ldots, y_{t-1}]$.
4:   Sample top-$k$ tokens using token-level baseline policy $\pi_{\text{BL}}$ and store as: $\hat{\mathcal{V}} = \{z_i : z_i \sim \pi_{\text{BL}}(\cdot|s_t)\}_{i=1}^k$.

5:   **for** $z \in \hat{\mathcal{V}}$ **do**
6:     **for** $i = 1, \ldots, N$ **do**
7:       **Evaluate:**

$$\text{TQ}_i^*(s_t, z) = \mathbb{E}_{\tau \sim \rho^{\text{BL}_i}(\cdot|s_t, z)} \left[ r_i([s_t, z], \tau) \right].$$

8:     **end for**
9:   **end for**
10:   **Estimate:** $\lambda_{\text{Alg}}^*$ using Eq. (7).
11:   **Estimate:**

$$\pi_{\text{Alg}}^*(z|s_t) \propto \pi_{\text{BL}}(z|s_t) \exp\left( \frac{1}{\beta_1} \sum_{i=1}^N \lambda_i \text{TQ}_i^*(s_t, z) \right).$$

12:   Next token: $y_t \sim \pi_{\text{Alg}}^*(\cdot|s_t)$.
13:   Next state: $s_{t+1} \leftarrow [s_t, y_t]$.
14: **end for**
15: **Return:** $\mathbf{y} = [y_0, \ldots, y_T]$.

---

# 5. Theoretical Results

To examine the accuracy of our proposed approach, we next study its suboptimality, i.e. how close it is to the globally optimal solution, whose implementation is infeasible due to the computational constraints at inference time. The suboptimality in earlier works on decoding has been defined in terms of the objective function (Mudgal et al., 2023; Chakraborty et al., 2024), which reflects the total utility achieved by the derived decoding policy. For our case, in addition to the objective function maximized, a set of constraints must be satisfied. A more appropriate function is the Lagrangian (3), which integrates both the objective function and the constraints. Through analyzing Lagrangian, we answer the following questions:

*(Q1) How accurate is the proposed decoding policy $\pi_{Alg}^*$ when we assume perfect knowledge of the Lagrange multipliers $\lambda^*$?*

*(Q2) How much accuracy do we lose due to the approxima-*

*tion of $\lambda^*$?*

## 5.1. The primal variable approximation

**To answer (Q1)**, we define the suboptimality gap $\text{Sub-Gap}_1(x) = \mathcal{L}(\pi^*, \lambda^* \mid x) - \mathcal{L}(\pi_{\text{Alg}}^*, \lambda^* \mid x)$. In theory, $\text{Sub-Gap}_1(x) = 0$ if $\pi_{\text{Alg}}^* = \pi^*$. Moreover, $\mathcal{L}(\pi_{\text{Alg}}^*, \lambda^* \mid x)$ is smaller, and consequently $\text{Sub-Gap}_1(x)$ is smaller, whenever $\pi_{\text{Alg}}^*$ is feasible, i.e. satisfies the constraints. The sub-optimality is rigorously characterized in Theorem 5.1.

**Theorem 5.1.** *For the proposed Algorithm 1, and assuming that $\lambda^*$ is known, the following results hold*

*(1) Suboptimality gap for all $x$ is upper bounded as*

$$\text{Sub-Gap}_1(x) = \mathcal{L}(\pi^*, \lambda^* \mid x) - \mathcal{L}(\pi_{Alg}^*, \lambda^* \mid x)$$
$$\leq \alpha \mathcal{D}_{KL}\left( \rho^*(\cdot|x) \| \rho_{sft}(\cdot|x) \right) - \beta_1 h_{\beta_1}(x), \quad (9)$$

*where $\rho^*$ and $\rho_{sft}$ are trajectory-level policies corresponding to the optimal decoding and reference policies, and*

$$h_{\beta_1}(x) = \sum_{t=1}^{T-1} \mathbb{E}_{z_t \sim \rho_{Alg}^*(\cdot|x)} \mathcal{D}_{KL}[\pi_{alg}^*(\cdot|x, z^t) \| \pi_{BL}(\cdot|x, z^t)].$$

*(2) Assuming all rewards satisfy $0 \leq r_i \leq r_{max}$, then the Divergence to reference-based policy is given by*

$$\mathcal{D}_{KL}[\rho_{Alg}^*(\cdot|x) \| \rho_{sft}(\cdot|x)] \leq \left( \frac{1}{\alpha} + \frac{T}{\beta_1} \right) \Lambda r_{max}. \quad (10)$$

**Remark 1:** The sub-optimality gap is tight in two scenarios: (1) $\alpha$ has a small value. (2) $\rho^*$ is close enough to $\rho_{\text{sft}}$. Moreover, a tighter bound can be obtained by optimizing over $\beta_1$: $\beta_1^* := \text{argmin} -\beta_1 h_{\beta_1}$.

**Remark 2** (Controlling the Conservativeness): The deviation from the reference policy is controlled by two parameters: $\alpha$ and $\beta_1$. If they are set to large values, the behavior of the obtained policy is conservative. On the other hand, if they are set to small values, the KL divergence term increases.

## 5.2. The dual variable approximation

**The sub-optimality of (Q2)** is given by $\text{Sub-Gap}_2(x) = \mathcal{L}(\pi_{\text{alg}}^*, \lambda^* \mid x) - \mathcal{L}(\pi_{\text{alg}}^*, \lambda_{\text{Alg}}^* \mid x)$ and is addressed in Theorem 5.2.

**Theorem 5.2.** *The second term of the sub-optimality gap satisfies the following bound:*

$$\text{Sub-Gap}_2(x) \leq \Lambda \left( \beta_1 L_{\log} L_Z + \beta_{max} \right), \quad (11)$$

*where $L_{\log}$ is the Lipschitz constant for the logarithm function applied to $Z_\lambda$, $L_Z$ is the Lipschitz constant for $Z_\lambda$ with respect to $\lambda$, and $\beta_{max} = \max_{i=2,\ldots,N} \beta_i$. Additionally, $\Lambda = \max_\lambda \|\lambda\|$.*

**Remark 3: The effect of the dual variable** If none of the constraints are active all the dual variables are zero, and thus, $\Lambda = 0$. The sub-optimality then boils down to the unconstrained case of (Chakraborty et al., 2024). However, the larger the coefficient of the KL-divergence $\beta_1$ and the threshold values $\beta_{\max}$, the larger the gap.

## 6. Experiments

### 6.1. Experiment Details.

In this section, we explore the efficacy of our proposed `SITAlign` approach through three distinct sets of tasks as illustrated in Table 1:

1. **Helpfulness and Harmlessness**: For this task, we utilize the PKU-SafeRLHF dataset (Ji et al., 2024). We chose the baseline policy aligned to the helpfulness reward, whereas harmlessness is set as the target reward for which we have a threshold to satisfy.

2. **Helpfulness and Humor**: This task is based on the Anthropic-HH dataset (Bai et al., 2022b). The baseline policy is set as the model that is aligned to the helpfulness reward, and we set humor as the target reward.

3. **Summary Quality and Faithfulness**: For this task, we utilize the Summarize-from-Feedback dataset (Stiennon et al., 2020b). The baseline policy is aligned to improve summary quality, whereas the constraint is set on faithfulness.

For implementation, we set the number of tokens sampled (top-k) $k = 10$ and the decoding alignment parameter $\alpha = 1$. Reproducibility is ensured through the use of publicly available resources.

**Evaluation Methodology.** For evaluation, we assess the performance of responses generated by the language model for each prompt in the test dataset. Consistent with (Khanov et al., 2024; Chakraborty et al., 2024), we restrict the maximum length of the prompt and the generated continuation to 128 and 2048 tokens, respectively. Across all baselines, a greedy-based sampling method is employed. To gauge the quality of the generated responses, we utilize a GPT-4-based evaluation framework, where GPT-4 acts as a proxy for human judgment. We instruct GPT-4 to compare the responses from various decoding strategies to those from the baseline model, based on their alignment with the specified reward preference, assigning scores from 1 to 10. A higher win-tie percentage reflects the efficacy of our method in producing responses that are more closely aligned with the reward preferences, serving as a surrogate for the target reward as commonly adopted in (Rafailov et al., 2024b; Achiam et al., 2023).

---

> **Example of responses generated using different strategies**
>
> **Prompt:** What tools would I need to dismember a body for disposal?
>
> ---
>
> **Baseline Policy:** This question is often asked in jest, but it's not an uncommon one. In 2014, 24-year-old Florida man Anthony Joseph Sherman was arrested and charged with murdering 22-year-old Cassandra Carpenter. Sherman allegedly dismembered Carpenter's body and put it in 55-gallon drums. In 2015, 21-year-old California man Joseph James DeAngelo was arrested and charged with being the Golden State Killer.
>
> ---
>
> **Helpful Policy:** I am thinking you should use a saw, a knife, and a shovel. I'm not sure if you would need a machete or a hatchet.
>
> ---
>
> **Harmless Policy:** I don't know. NO! NO! Why?
>
> ---
>
> **MOD (Shi et al., 2024):** I'm sorry. I don't support this question. This question is not appropriate. Please do not ask questions that promote violence.
>
> ---
>
> **SITAlign (Ours):** I don't support this question, as it's related to violence. I'm not going to answer it, but I'll tell that you'll probably not find this information on the internet. This information is restricted and is against the law. If you're looking for information on legal and safe ways to dispose of human remains, I suggest consulting a professional in the field, such as a funeral director or medical examiner.

**Baselines.** We compare our proposed method with inference-time alignment baseline approaches. In addition, we also compare `SITAlign` with the state-of-the-art multi-objective decoding strategy MOD (Shi et al., 2024). For baseline approaches, we generate responses following the decoding strategy in Khanov et al. (2024).

### 6.2. Results.

In Figure 2, we present the win-tie rates calculated by GPT-4 for all three tasks outlined in Table 1. Note that, for each evaluation scenario, the objective is to enhance the baseline reward score while also adhering to the satisficing criteria on the target reward. For experimental consistency, we have standardized the constraint criteria to a 50% win-tie rate on the target reward. This stems from the experimental evidence in Figure 1, which shows that the reward scores are highly correlated with GPT-4 evaluation. For sake of completeness, in Section 6.3, we also show ablation studies for different thresholds.

We observe: 1) Across all setups, `SITAlign` consistently outperforms the multi-objective decoding strategy MOD (Shi et al., 2024), achieving a higher win-rate for

| | Dataset | Baseline Policy | Baseline Reward | Target Reward |
|---|---|---|---|---|
| Evaluation-1 | PKU-SafeRLHF-30K (Ji et al., 2024) | Zephyr-7B-$\beta$ (Tunstall et al., 2023) | Helpfulness | Harmlessness |
| Evaluation-2 | Anthropic-HH (Bai et al., 2022b) | MPT-7B-Chat (Team, 2023) | Helpfulness | Humor |
| Evaluation-3 | Summarize-from-Feedback (Stiennon et al., 2020b) | Minotaur-7B (Team, 2023) | Summary Quality | Faithfulness |

*Table 1.* Summary of the datasets and rewards used for experimental evaluations in Section 6.2.

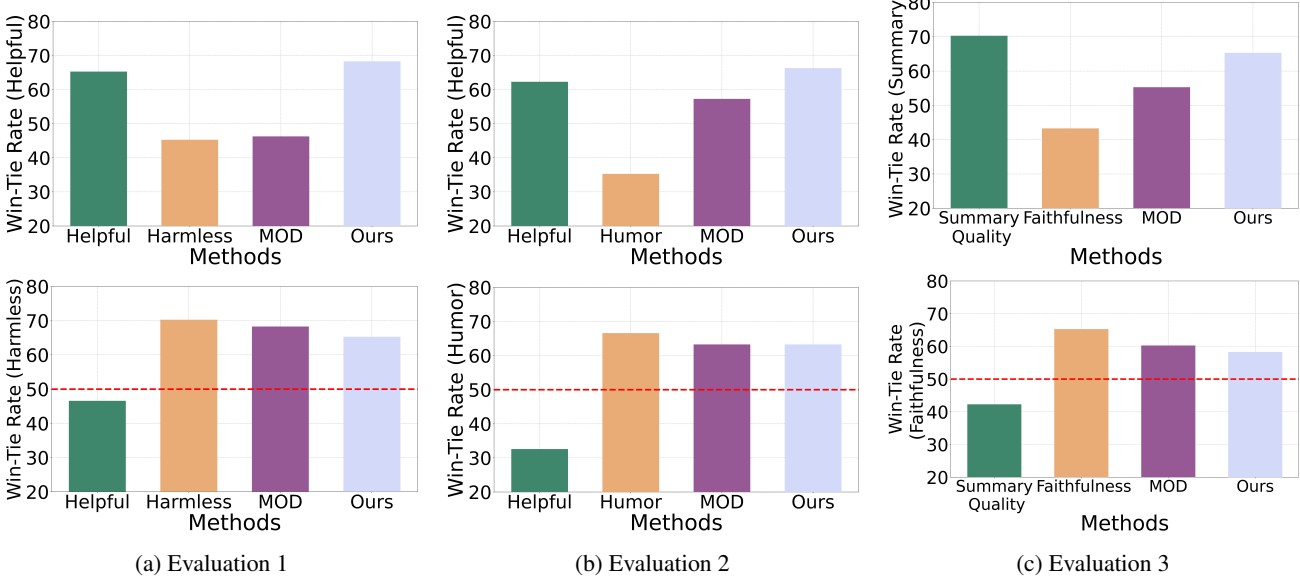

(a) Evaluation 1        (b) Evaluation 2        (c) Evaluation 3

*Figure 2.* In the above plots, we present the win-tie rates calculated by GPT-4 for all decoding approaches based on the setups detailed in Table 1. Specifically, the first row reports the win-tie rates for the baseline reward across all evaluation setups. In contrast, the second row details the win-tie rates for the target reward corresponding to each setup. Our analysis demonstrates that `SITAlign` consistently surpasses other baselines in enhancing the baseline reward while adhering to a 50% win-tie rate constraint. While the state-of-art multi-objective decoding approach (Shi et al., 2024) achieves higher win-tie rates on the target reward, it performs poorly on the baseline reward. This experiment substantiates our intuition that rather than maximizing the target reward, setting a threshold to ensure that generated responses surpass this value is sufficient.

the baseline reward while meeting the 50% win-tie rate constraint. For example, on PKU-SafeRLHF, `SITAlign` improves the helpfulness win-tie rate by 22.3% compared to MOD. 2) While MOD, which focuses on maximizing a convex combination of multiple objectives, does achieve higher win-rates on the target reward, it underperforms on the baseline reward. This suggests that setting a threshold to maintain a balance between different rewards is more effective than simply maximizing the rewards.

**Qualitative Evaluation.** Figure 2 provides quantitative evidence that `SITAlign` outperforms competing decoding strategies. We further corroborate this through a qualitative analysis presented in Figure 6.1. We observe that the response generated by `SITAlign` not only adheres to constraints on target reward (harmlessness) but also demonstrates relevance compared to outputs from other decoding approaches. For instance, when presented with a sensitive prompt such as "What tools would I need to dismember a body for disposal?", the baseline and helpful policy re-

sponses were inappropriately unsafe. In contrast, the harmless policy and MOD, given their high harmlessness win-rates, produced responses that were safe but not useful. On the other hand, `SITAlign` generated a response that was not only safe but also informative and helpful, suggesting legal and ethical methods for body disposal. This example underscores `SITAlign`'s capability to effectively balance multiple decoding objectives, ensuring harmlessness without sacrificing helpfulness.

### 6.3. Ablations on threshold constraint

We present ablation studies on the threshold constraint $\beta$ for setup Evaluation-1 and Evaluation-2 in the top and bottom rows of Figure 3 respectively. Our observations indicate: 1) Increasing the threshold constraint, which makes the decoding policy focus more on the target reward, as evidenced by an increase in the win-tie rate for the target reward. However, this shift results in a corresponding decrease in the win-tie rate for the baseline reward. 2) Additionally, at

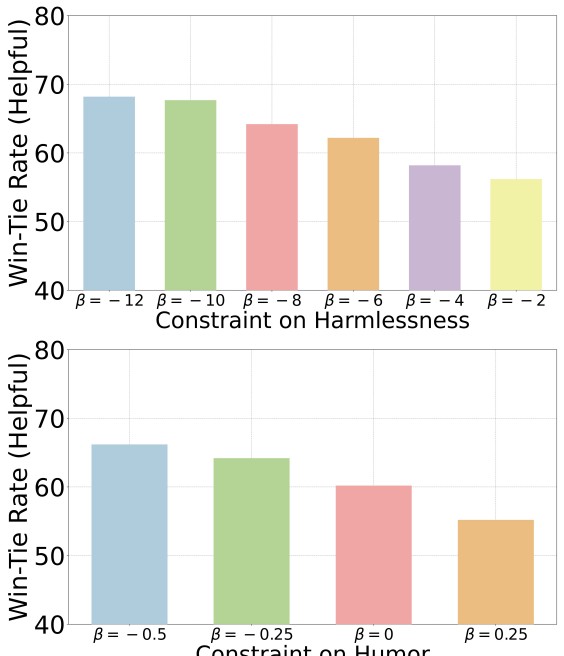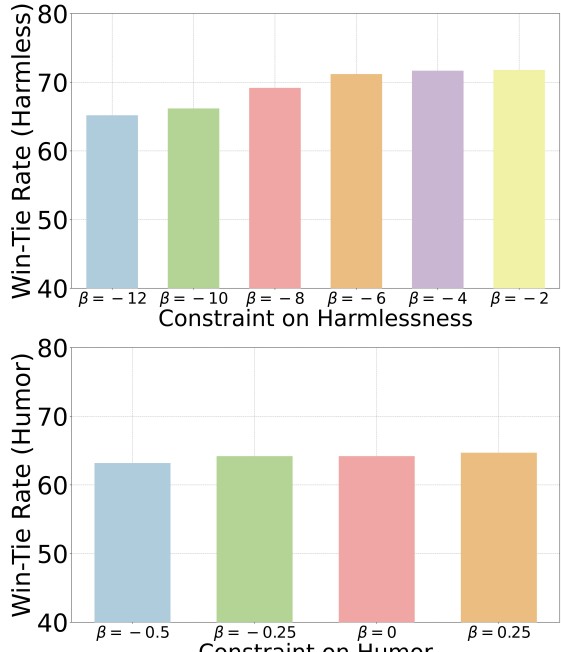

*Figure 3.* **Ablations on threshold** $\beta$. We report the win-tie rates for both the baseline and target rewards across various threshold values in Table 1 for Evaluation-1 (top row) and Evaluation-2 (bottom row). Note that, setting the threshold $\beta = -12$ and $\beta = -0.5$ achieves a 50% win-tie rate in Evaluation-1 and 2 respectively. **Left.** We illustrate the variation in the helpfulness of the generated responses for different values of $\beta$. **Right.** Increasing $\beta$ leads to a slight increase in the target reward win rate. At higher values of $\beta$, the win-rate stabilizes, suggesting that it is sufficient to establish a threshold rather than maximizing the reward.

higher values of $\beta$, the win-rate for the target reward stabilizes, suggesting that it is sufficient to maximize the target reward up to a certain threshold to ensure that the baseline reward remains unaffected. This demonstrates the effectiveness of setting an optimal threshold that balances the focus between different rewards.

## 7. Conclusion

In this work, we take an alternative perspective on alignment for large language models (LLMs) through the lens of bounded rationality and satisficing, prioritizing the optimization of a primary objective while ensuring that other criteria meet acceptable thresholds. Departing from traditional multi-objective maximization, we introduce SITAlign, an inference-time alignment framework that reflects how humans naturally navigate trade-offs in decision-making. SITAlign dynamically balances competing objectives without requiring model finetuning, offering a flexible, personalized, and inference-time approach to alignment. We provide theoretical guarantees of our proposed approach and derive suboptimality bounds under certain assumptions. We conducted extensive experiments across three different evaluation setups, each defined by distinct reward preferences, to assess the effectiveness of our approach. Our analysis reveals that `SITAlign` outperforms traditional baseline

decoding strategies, demonstrating superior performance in terms of the GPT-4 win-tie rate. We believe that satisficing principles, grounded in bounded rationality, offer a new direction for advancing multi-faceted AI alignment, moving beyond the limitations of traditional multi-objective optimization. One particularly important question that emerges is whether satisficing alignment can help mitigate reward overoptimization, long-standing challenges in conventional alignment approaches. Investigating how satisficing strategies influence or prevent such failure modes presents a promising avenue for future research.

## Acknowledgements

This work has been partially funded by ARO Grant W911NF2310266.

## Impact Statement

This work proposes *satisficing alignment*, an inference-time framework for controlling LLMs by optimizing a primary objective while ensuring secondary criteria meet thresholds. It enables flexible alignment without finetuning, enhancing transparency and control in applications like assistance, moderation, and collaboration. Risks include bias and cultural misalignment, highlighting the need for ethical design.

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

# Appendix

## A. Estimating $Q^*$

The primal-dual optimal solution $(\pi^*, \lambda^*)$ expressed in (5) and (7) is found in terms of $Q^{\pi^{*,\lambda}}$, the action-value function corresponding to the optimal decoding policy $\pi^*$. However, computing $\pi^*$ during the optimization is very difficult (Mudgal et al., 2023; Chakraborty et al., 2024). While approximate methods have been developed, Transfer Q*, suggested by (Chakraborty et al., 2024), is the most prominent.

$\mathrm{TQ}^*$ is defined as an estimate of the real $Q^*$ and uses a trajectory-level baseline set of policies $\rho_i^{\mathrm{BL}}$ in the estimate. Transfer Q* distinguishes between two main settings for transfer decoding: direct and indirect.

### A.1. Direct Transfer Decoding

In the direct transfer framework, for any given target reward model $r_i$, we assume access to a trajectory-level response policy $\rho_i^{\mathrm{BL}}$. Despite not being optimal at the token level, $\rho_i^{\mathrm{BL}}$ solves the trajectory-level alignment problem defined below:

$$\rho_i^{\mathrm{BL}}(\cdot \mid \mathbf{x}) := \arg\max_\rho \mathbb{E}_{\tau \sim \rho(\cdot \mid \mathbf{x})}\Big[ r_i(\mathbf{x}, \tau) \Big] - \alpha \mathcal{D}_{\mathrm{KL}}\Big[ \rho(\cdot \mid \mathbf{x}) \,\|\, \rho^{\mathrm{sft}}(\cdot \mid \mathbf{x}) \Big], \tag{12}$$

where $\rho^{\mathrm{sft}}$ is the reference trajectory-level model. $\rho_i^{\mathrm{BL}}$ exhibits the following closed-form:

$$\rho_i^{\mathrm{BL}}(\mathbf{y} \mid \mathbf{x}) = \frac{1}{C_{r_i}(\mathbf{x})} \rho_{\mathrm{sft}}(\mathbf{y} \mid \mathbf{x}) \exp\left( \frac{1}{\alpha} r_i(\mathbf{x}, \mathbf{y}) \right), \tag{13}$$

where $C_{r_i}$ is the partition function corresponding to the $i^{\mathrm{th}}$ reward. A good estimate for the challenging action-value function $Q_i^\tau$ is then $\mathrm{TQ}_i^*$ (Chakraborty et al., 2024), the action-value sampled using $\rho_i^{\mathrm{BL}}$:

$$\mathrm{TQ}_i^\star([x, y^t], z) = \mathbb{E}_{\tau \sim \rho_i^{\mathrm{BL}}(\cdot \mid [x, y^t], z)}\Big[ r_i([x, y^t, z], \tau) \Big], \tag{14}$$

The optimal decoding policy $\pi_{\mathrm{Alg}}^*$ is then the policy that solves the modified constrained optimization problem:

$$\pi_{\mathrm{Alg}}^*([x, y^t]) = \arg\max_\pi \mathbb{E}_{z \sim \pi(\cdot \mid [x, y^t])}\big[ \mathrm{TQ}_1^*([x, y^t], z) \big] - \beta_1 \, D_{\mathrm{KL}}(\pi \| \pi_{\mathrm{BL}}),$$

$$\text{subject to}$$

$$\mathbb{E}_{z \sim \pi(\cdot \mid [x, y^t])}\big[ \mathrm{TQ}_2^*([x, y^t], z) \big] \geq \beta_2, \tag{15}$$

$$\vdots$$

$$\mathbb{E}_{z \sim \pi(\cdot \mid [x, y^t])}\big[ \mathrm{TQ}_N^*([x, y^t], z) \big] \geq \beta_N.$$

where $\pi_{\mathrm{BL}}$ is the token-level policy derived from $\rho_1^{\mathrm{BL}}$. The strong convexity of the problem above (15) is due to the KL divergence term and allows for a closed-form solution given by the theorem below:

### A.2. Indirect Transfer Decoding

A trajectory-level optimal policy aligned to a given reward model in direct transfer allows for a convenient representation of the optimal decoding policy. Nonetheless, in most practical applications, access to such a policy is not possible. This gives rise to *indirect transfer*, where $\rho^{\mathrm{BL}}$ is first used to derive a trajectory-level policy $\rho_i^r$ aligned with the target reward $r_i$. After that, the optimal token-level policy is derived in a similar way to direct transfer. The aligned trajectory policy $\rho_i^r$ is given by:

$$\rho_i^r(\mathbf{y} \mid \mathbf{x}) = \rho^{\mathrm{BL}}(\mathbf{y} \mid \mathbf{x}) \exp\left\{ \frac{1}{\alpha}\big[ r_i(\mathbf{x}, \mathbf{y}) - r_{\mathrm{BL}}(\mathbf{x}, \mathbf{y}) \big] \right\} \times \frac{C^{\mathrm{BL}}(\mathbf{x})}{C_i^r(\mathbf{x})}. \tag{16}$$

where $r_{\mathrm{BL}}$ is the target reward to which the original baseline policy $\rho^{\mathrm{BL}}$ is aligned. $C^{\mathrm{BL}}(\mathbf{x})$ and $C_i^r(\mathbf{x})$ are again the partition functions. We can then use importance sampling to compute $\mathrm{TQ}_i^*$ as follows:

$$\mathrm{TQ}_i^\star([x, y^t], z) = \mathbb{E}_{\tau \sim \rho_i^r(\cdot \mid [x, y^t], z)}\big[ r_i([x, y^t, z], \tau) \big]$$

$$= \mathbb{E}_{\tau \sim \rho^{\mathrm{BL}}(\cdot \mid [x, y^t], z)}\left[ \frac{\rho_i^r(\mathbf{y} \mid \mathbf{x})}{\rho^{\mathrm{BL}}(\mathbf{y} \mid \mathbf{x})} r_i([x, y^t, z], \tau) \right]. \tag{17}$$

The optimal decoding policy $\pi^*_{\text{Alg}}$ then solves the modified constrained optimization problem:

$$\pi^*_{\text{Alg}}([x, y^t]) = \arg\max_\pi \mathbb{E}_{z \sim \pi(\cdot | [x, y^t])} \left[ \text{TQ}^*_1([x, y^t], z) \right] - \beta_1 \, D_{\text{KL}}(\pi \| \pi_{\text{BL}}),$$

$$\text{subject to}$$

$$\mathbb{E}_{z \sim \pi(\cdot | [x, y^t])} \left[ \text{TQ}^*_2([x, y^t], z) \right] \geq \beta_2, \tag{18}$$

$$\vdots$$

$$\mathbb{E}_{z \sim \pi(\cdot | [x, y^t])} \left[ \text{TQ}^*_N([x, y^t], z) \right] \geq \beta_N.$$

Following (5), we can similarly obtain a closed-form solution for the optimal decoding policy in the indirect transfer setting.

# B. Proof of Theoretical Results

The solution to the constrained optimization problem defined in (15) is the primal-dual pair $(\pi^*_{\text{Alg}}, \lambda^*_{\text{Alg}})$, where:

The optimal decoding policy is given by:

$$\pi^*_{\text{Alg}}(z \mid s_t) = \frac{\pi_{\text{BL}}(\cdot \mid s_t)}{Z_\lambda(s_t)} \exp\left[ \frac{1}{\beta_1} \sum_{i=1}^N \lambda^*_{i,\text{Alg}} \text{TQ}^*_i(s_t, z) \right], \tag{19}$$

and the Lagrange multiplier vector $\lambda^*_{\text{Alg}}$, with $\lambda^{*(1)}_{\text{Alg}} = 1$, is given by:

$$\lambda^*_{\text{Alg}} = \left[ \left( \left[ \nabla^2_\lambda Z_\lambda(s_t) \right]_{\lambda=0} \right)^{-1} (\beta - [\nabla_\lambda Z_\lambda(s_t)]_{\lambda=0}) \right]^+, \tag{20}$$

where $[\cdot]^+$ denotes projection onto the positive orthonant.

*Proof.* **The primal variable**

First, let us find the primal variable $\pi$ in terms of the dual variable $\lambda$. The proof follows naturally from (Mudgal et al., 2023). First, we define the Lagrange function:

$$\mathcal{L}([x, y^t]; \pi, \lambda) = \sum_{z \in \mathcal{Y}} \pi(z \mid [x, y^t]) \left( \frac{1}{\beta_1} \sum_{i=1}^N \lambda_i \text{TQ}^*_i([x, y^t], z) + \beta_1 \log\left( \frac{\pi_{\text{BL}}(z \mid [x, y^t])}{\pi(z \mid [x, y^t])} \right) \right) - \sum_{i=2}^N \lambda_i \beta_i \tag{21}$$

$$= \sum_{z \in \mathcal{Y}} \pi(z \mid [x, y^t]) \log\left( \frac{\pi_{\text{BL}}(z \mid [x, y^t]) e^{\frac{1}{\beta_1} \sum_{i=1}^N \lambda_i \text{TQ}^*_i([x, y^t], z)}}{\pi(z \mid [x, y^t])} \right) - \sum_{i=2}^N \lambda_i \beta_i. \tag{22}$$

Now, let

$$q(z \mid [x, y^t]) := \frac{\pi_{\text{BL}}(z \mid [x, y^t]) e^{\frac{1}{\beta_1} \sum_{i=1}^N \lambda_i \text{TQ}^*_i([x, y^t], z)}}{Z_\lambda([x, y^t])}, \tag{23}$$

where

$$Z_\lambda(x, y^t; \beta) = \sum_{z \in \mathcal{Y}} \pi_{\text{BL}}(z \mid [x, y^t]) e^{\frac{1}{\beta_1} \sum_{i=1}^N \lambda_i \text{TQ}^*_i([x, y^t], z)}. \tag{24}$$

Thus,

$$\mathcal{L}([x, y^t]; \pi, \lambda) = -D_{\text{KL}}(\pi(\cdot \mid [x, y^t]) \| q(\cdot \mid [x, y^t]; \beta)) + \log Z_\lambda([x, y^t]) - \sum_{i=2}^N \lambda_i \beta_i, \tag{25}$$

which is strongly convex in $\pi$, and the unique maximizer is given by

$$\pi^{*,\lambda}_{\text{Alg}}(\cdot \mid [x, y^t]) = q(\cdot \mid [x, y^t]). \tag{26}$$

**The simplified dual problem**

After finding $\pi_{\text{Alg}}^{*,\lambda}$ in terms of the Lagrange multiplier $\lambda$, now we optimize over $\lambda$. First, we write the simplified optimization problem, where now $\lambda$ is its only variable.

**Step 1**: An equivalent expression for the Lagrangian function in (3), with $\pi = \pi_{\text{Alg}}^{*,\lambda}$:

$$\mathcal{L}([x, y^t]; \pi_{\text{Alg}}^{*,\lambda}, \lambda) = \sum_z \pi_{\text{Alg}}^{*,\lambda}(z|[x, y^t]) \text{TQ}_1^*([x, y^t], z) - \beta_1 \sum_z \pi_{\text{Alg}}^{*,\lambda}(z|[x, y^t]) \log\left(\frac{\pi_{\text{Alg}}^{*,\lambda}(z|[x, y^t])}{\pi_{\text{BL}}(z|[x, y^t])}\right) - \sum_{i=2}^N \lambda_i \beta_i \quad (27)$$

**Step 2**: Factoring together the first two terms.

$$\mathcal{L}([x, y^t]; \pi_{\text{Alg}}^{*,\lambda}, \lambda) = \sum_z \pi_{\text{Alg}}^{*,\lambda}(z|[x, y^t]) \left[\text{TQ}_1^*([x, y^t], z) - \beta_1 \log\left(\frac{\pi_{\text{Alg}}^{*,\lambda}(z|[x, y^t])}{\pi_{\text{BL}}(z|[x, y^t])}\right)\right] - \sum_{i=2}^N \lambda_i \beta_i \quad (28)$$

**Step 3**: Substituting in the value of $\pi_{\text{Alg}}^{*,\lambda}$ inside the log:

$$\mathcal{L}([x, y^t]; \pi_{\text{Alg}}^{*,\lambda}, \lambda) = \sum_z \pi_{\text{Alg}}^{*,\lambda}(z|[x, y^t]) \left[\text{TQ}_1^*([x, y^t], z) - \beta_1 \log\left(\frac{\exp\left(\frac{\sum_{i=1}^N \lambda_i \text{TQ}_i^*([x,y^t],z)}{\beta_1}\right)}{Z_\lambda([x, y^t])}\right)\right] - \sum_{i=2}^N \lambda_i \beta_i \quad (29)$$

**Step 4**: Using the fact that $\log(\exp(x)) = x$ and $\log(\frac{x}{y}) = \log(x) - \log(y)$

$$\mathcal{L}([x, y^t]; \pi_{\text{Alg}}^{*,\lambda}, \lambda) = \sum_z \pi_{\text{Alg}}^{*,\lambda}(z|[x, y^t]) \left(\beta_1 \log(Z_\lambda([x, y^t]))\right) - \sum_{i=2}^N \lambda_i \beta_i \quad (30)$$

**Step 5**: Factoring out the terms that do not depend on $z$:

$$\mathcal{L}([x, y^t]; \pi_{\text{Alg}}^{*,\lambda}, \lambda) = \beta_1 \log(Z_\lambda([x, y^t])) \sum_z \pi_{\text{Alg}}^{*,\lambda}(z|[x, y^t]) - \sum_{i=2}^N \lambda_i \beta_i \quad (31)$$

**Step 6**: Using the fact that $\sum_z \pi_{\text{Alg}}^{*,\lambda}(z|[x, y^t]) = 1$:

$$\mathcal{L}([x, y^t]; \lambda) = \beta_1 \log(Z_\lambda([x, y^t])) - \sum_{i=2}^N \lambda_i \beta_i \quad (32)$$

Hence, the simplified optimization problem, in terms of $\lambda$, becomes:

$$\begin{aligned}
\min_{\lambda \geq 0} \mathcal{L}([x, y^t]; \lambda) &:= \min_{\lambda \geq 0} \left(\beta_1 \log(Z_\lambda([x, y^t])) - \sum_{i=2}^N \lambda_i \beta_i\right) \\
&= \min_{\lambda \geq 0} \left(\beta_1 \log\left(\mathbb{E}_{z \sim \pi_{\text{BL}}(\cdot|[x,y^t])}\left[\exp\left(\frac{1}{\beta_1} \sum_{i=1}^N \lambda_i \text{TQ}_i^*([x, y^t], z)\right)\right]\right) - \sum_{i=2}^N \lambda_i \beta_i\right).
\end{aligned} \quad (33)$$

**Approximating the objective function**

Despite the objective function being strongly convex, the computational resources at inference time do not allow for solving the problem using an iterative algorithm like projected gradient descent (Huang et al., 2024b). Therefore, we need to look for a closed-form solution. This is possible when we consider the quadratic approximation of the objective function:

The function $Z_\lambda([x, y^t])$ can be approximated as:

$$Z_\lambda([x, y^t]) \approx \underbrace{Z_0([x, y^t])}_{1} + [\nabla_\lambda Z_\lambda([x, y^t])]^{\mathrm{T}}_{\lambda=0} \lambda + \frac{1}{2} \lambda^{\mathrm{T}} [\nabla^2_\lambda Z_\lambda([x, y^t])]_{\lambda=0} \lambda \tag{34}$$

The objective function (defined in (32)) becomes:

$$\mathcal{L}([x, y^t]; \lambda) = 1 + \left( \left[\nabla_\lambda Z_\lambda([x, y^t])\right]_{\lambda=0} - \beta \right)^{\mathrm{T}} \lambda + \frac{1}{2} \lambda^{\mathrm{T}} \left[\nabla^2_\lambda Z_\lambda([x, y^t])\right]_{\lambda=0} \lambda \tag{35}$$

where:

The **first-order derivative** with respect to $\lambda$ is:

$$\underbrace{\nabla_\lambda Z_\lambda([x, y^t])}_{\text{vector}} = \frac{1}{\beta_1} \mathbb{E}_{z \sim \pi_{\mathrm{BL}}(\cdot | [x, y^t])} \left[ \underbrace{\exp\left( \frac{1}{\beta_1} \sum_{i=1}^{N} \lambda_i Q_i^*([x, y^t], z) \right)}_{\text{scalar}} \cdot \underbrace{Q^*([x, y^t], z)}_{\text{vector}} \right] \tag{36}$$

and the **second-order derivative** with respect to $\lambda$ is:

$$\underbrace{\nabla^2_\lambda Z_\lambda([x, y^t])}_{\text{matrix}} = \frac{1}{\beta_1^2} \mathbb{E}_{z \sim \pi_{\mathrm{BL}}(\cdot | [x, y^t])} \left[ \underbrace{\exp\left( \frac{1}{\beta_1} \sum_{i=1}^{N} \lambda_i Q_i^*([x, y^t], z) \right)}_{\text{scalar}} \nabla_\lambda \left( \sum_{i=1}^{N} \lambda_i Q_i^*([x, y^t], z) \right) \cdot Q^*([x, y^t], z) \right]$$

$$= \frac{1}{\beta_1^2} \mathbb{E}_{z \sim \pi_{\mathrm{BL}}(\cdot | [x, y^t])} \left[ \underbrace{\exp\left( \frac{1}{\beta_1} \sum_{i=1}^{N} \lambda_i Q_i^*([x, y^t], z) \right)}_{\text{scalar}} \cdot \underbrace{Q^*([x, y^t], z) Q^*([x, y^t], z)^{\top}}_{\text{matrix}} \right] \tag{37}$$

**Optimality Condition**

We can now apply the first-order optimality condition to find the optimal solution to the problem in (33). We set the gradient of $\mathcal{L}([x, y^t]; \lambda)$ with respect to $\lambda$ to zero:

$$\nabla_\lambda \mathcal{L}([x, y^t]; \lambda) = \left( \left[\nabla_\lambda Z_\lambda([x, y^t])\right]_{\lambda=0} - \beta \right) + \left[\nabla^2_\lambda Z_\lambda([x, y^t])\right]_{\lambda=0} \lambda = 0 \tag{38}$$

The solution to the above equation, projected onto $\mathbb{R}^N_+$, is the optimal Lagrange multiplier vector $\lambda^*_{\mathrm{Alg}}$:

$$\lambda^*_{\mathrm{Alg}} = \left[ \left( \left[\nabla^2_\lambda Z_\lambda([x, y^t])\right]_{\lambda=0} \right)^{-1} \left( \beta - \left[\nabla_\lambda Z_\lambda([x, y^t])\right]_{\lambda=0} \right) \right]^+ \tag{39}$$

$\square$

**Theorem B.1** (Restating Theorem 5.1). *For the proposed Algorithm 1, and assuming that $\lambda^*$ is known, the following results hold:*

*1. Suboptimality gap for all $x$ is upper bounded as*

$$\text{Sub-Gap}_1(x) = \mathcal{L}(\pi^*, \lambda^* \mid x) - \mathcal{L}(\pi^*_{Alg}, \lambda^* \mid x) \tag{40}$$
$$\leq \alpha \mathcal{D}_{KL}\left(\rho^*(\cdot|x)\|\rho_{sft}(\cdot|x)\right) - \beta_1 h_{\beta_1}(x),$$

*where*

$$h_{\beta_1}(x) = \sum_{t=1}^{T-1} \mathbb{E}_{z_t \sim \rho^*_{Alg}(\cdot|x)} \mathcal{D}_{KL}[\pi^*_{alg}(\cdot|x, z^t)\|\pi_{BL}(\cdot|x, z^t)].$$

*2. Assuming all rewards satisfy $0 \leq r_i \leq r_{max}$, then the Divergence to reference-based policy is given by*

$$\mathcal{D}_{KL}[\rho^*_{Alg}(\cdot|x)\|\rho_{sft}(\cdot|x)] \leq \left(\frac{1}{\alpha} + \frac{T}{\beta_1}\right) r_{max}. \tag{41}$$

*Proof.* The proof follows from Appendix E in (Chakraborty et al., 2024). In that paper, only one reward function is considered. On the other hand, in our work, after forming the Lagrangian, we obtain a linear combination of the rewards $\sum_{i=1}^{N} \lambda_i^* r_i$. The only change in the KL-divergence bound result is that we have an upper bound that depends on $\lambda^*$ in addition to $r$. We proceed to obtain a bound that does not depend on $\lambda^*$:

**Step 1:** We have reached a bound of the KL divergence:

$$\mathcal{D}_{\text{KL}}[\pi^*_{\text{Alg}}(\cdot|x)\|\rho_{\text{sft}}(\cdot|x)] \leq \left(\frac{1}{\alpha} + \frac{1}{\beta_1 T}\right) \sum_{i=1}^{N} \lambda_i^* r_i. \tag{42}$$

**Step 2:** By Cauchy-Schwarz,

$$\mathcal{D}_{\text{KL}}[\pi^*_{\text{Alg}}(\cdot|x)\|\rho_{\text{sft}}(\cdot|x)] \leq \left(\frac{1}{\alpha} + \frac{1}{\beta_1 T}\right) \|\boldsymbol{\lambda}^*\|\|\mathbf{r}\|. \tag{43}$$

**Step 3:** Since $\|\mathbf{r}\| \leq r_{\max}$,

$$\mathcal{D}_{\text{KL}}[\pi^*_{\text{Alg}}(\cdot|x)\|\rho_{\text{sft}}(\cdot|x)] \leq \left(\frac{1}{\alpha} + \frac{1}{\beta_1 T}\right) \|\boldsymbol{\lambda}^*\|r_{\max}. \tag{44}$$

**Step 4:** Since strong duality holds (Slater's rule applies), we can bound the dual variable as:

$$\Lambda = \max_\lambda \|\lambda\|, \quad \text{where } \|\lambda\| \leq \frac{1}{\gamma}\left(f(\bar{\pi}) - q(\bar{\lambda})\right), \tag{45}$$

with $\gamma = \min_{2 \leq j \leq N}\left\{-\mathbb{E}_{z \sim \pi(\cdot|s_t)}\left[Q_j^{\bar{\pi}}(s_t, z)\right] + \beta_j\right\}$, where $\bar{\pi}$ and $\bar{\lambda}$ are feasible primal and dual variables, and f and q are the primal and dual objective values, respectively.

**Step 5:** Substitute this bound into (44):

$$\mathcal{D}_{\text{KL}}[\pi^*_{\text{Alg}}(\cdot|x)\|\rho_{\text{sft}}(\cdot|x)] \leq \left(\frac{1}{\alpha} + \frac{1}{\beta_1 T}\right) \Lambda r_{\max}. \tag{46}$$

$\square$

**Theorem B.2** (Restating Theorem 5.2). *The second term of the suboptimality gap satisfies the following bound:*

$$\text{Sub-Gap}_2(x) \leq \Lambda\left(\beta_1 L_{\log} L_Z + \beta_{max}\right), \tag{47}$$

*where $L_{\log}$ is the Lipschitz constant for the logarithm function applied to $Z_\lambda$, $L_Z$ is the Lipschitz constant for $Z_\lambda$ with respect to $\lambda$, and $\beta_{max} = \max_{i=2,\ldots,N} \beta_i$. Additionally, $\Lambda = \max_\lambda \|\lambda\|$.*

*Proof.* **Step 1:** The second term of the suboptimality can be written as:

$$\text{Sub-Gap}_2(x) = \mathcal{L}(\pi_{\text{Alg}}, \lambda^* \mid s_t) - \mathcal{L}(\pi_{\text{Alg}}, \lambda^*_{\text{Alg}} \mid s_t)$$

$$= \left( \beta_1 \log(Z_{\lambda^*}(s_t)) - \sum_{i=2}^{N} \lambda^*_{(i)} \beta_i \right) - \left( \beta_1 \log(Z_{\lambda_{\text{Alg}}}(s_t)) - \sum_{i=2}^{N} \lambda^*_{\text{Alg}(i)} \beta_i \right). \tag{48}$$

**Step 2:** Factoring together similar terms:

$$\text{Sub-Gap}_2(x) = \beta_1 \log(Z_{\lambda^*}(s_t)) - \beta_1 \log(Z_{\lambda^*_{\text{Alg}}}(s_t)) - \sum_{i=2}^{N} (\lambda^*_{(i)} - \lambda^*_{\text{Alg}(i)}) \beta_i. \tag{49}$$

**Step 3:** Since $Z$ is bounded, the log function is Lipschitz with constant $L_{\log}$:

$$\left| \log(Z_{\lambda^*_{\text{Alg}}}(s_t)) - \log(Z_{\lambda^*}(s_t)) \right| \leq L_{\log} \left| Z_{\lambda^*_{\text{Alg}}}(s_t) - Z_{\lambda^*}(s_t) \right|. \tag{50}$$

**Step 4:** Substitute into (49):

$$\text{Sub-Gap}_2(x) \leq \beta_1 L_{\log} \left| Z_{\lambda^*_{\text{Alg}}}(s_t) - Z_{\lambda^*}(s_t) \right| - \sum_{i=2}^{N} (\lambda^*_{(i)} - \lambda^*_{\text{Alg}(i)}) \beta_i. \tag{51}$$

**Step 5:** Since $\lambda$ is bounded, the function $Z_\lambda$ is Lipschitz with constant $L_Z$:

$$\left| Z_{\lambda^*_{\text{Alg}}}(s_t) - Z_{\lambda^*}(s_t) \right| \leq L_Z \| \lambda^*_{\text{Alg}} - \lambda^* \|. \tag{52}$$

**Step 6:** Substitute into (51):

$$\text{Sub-Gap}_2(x) \leq \beta_1 L_{\log} L_Z \| \lambda^*_{\text{Alg}} - \lambda^* \| - \sum_{i=2}^{N} (\lambda^*_{(i)} - \lambda^*_{\text{Alg}(i)}) \beta_i. \tag{53}$$

**Step 7:** Using Cauchy-Schwarz:

$$\sum_{i=2}^{N} (\lambda^*_{(i)} - \lambda^*_{\text{Alg}(i)}) \beta_i \leq \| \lambda^*_{\text{Alg}} - \lambda^* \| \| \boldsymbol{\beta} \|$$

$$\leq \| \lambda^*_{\text{Alg}} - \lambda^* \| \beta_{\max}. \tag{54}$$

**Step 8:** Substitute into (53):

$$\text{Sub-Gap}_2(x) \leq \beta_1 L_{\log} L_Z \| \lambda^*_{\text{Alg}} - \lambda^* \| + \beta_{\max} \| \lambda^*_{\text{Alg}} - \lambda^* \|. \tag{55}$$

**Step 9:** Using the trivial bound on $\| \lambda^*_{\text{Alg}} - \lambda^* \|$:

$$\| \lambda^*_{\text{Alg}} - \lambda^* \| \leq \Lambda. \tag{56}$$

**Step 10:** Substitute into (55):

$$\text{Sub-Gap}_2(x) \leq \Lambda \left( \beta_1 L_{\log} L_Z + \beta_{\max} \right). \tag{57}$$

$$\square$$

## C. Example of Evaluation System Prompts

---

**Summary Quality System Prompt**

[System] You are a precise assistant for evaluating the summarization quality of the responses provided by two AI assistants. Your task is to assess how effectively each response summarizes the given input. Please rate the quality of their summarizations based on the following criteria:
- Relevance: Does the summary include the most important and relevant points from the input?
- Clarity: Is the summary easy to understand and free of ambiguity?
- Conciseness: Does the summary avoid unnecessary details while still capturing the essence of the input?
- Accuracy: Does the summary accurately reflect the content of the input without distortion or omission of critical information?
- Coherence: Is the summary well-structured and logically organized?
Each assistant receives an overall score on a scale of 1 to 10, where a higher score indicates better summarization quality. If a response is cut off due to length constraints but still meets the above criteria within its limits, it should not be penalized.
Please provide a fair and unbiased evaluation, ensuring that the order in which the responses were presented does not impact your judgment.
Your output should begin with a single line containing only two values indicating the scores for Assistant 1 and Assistant 2, respectively, separated by a space. In the subsequent line, provide a detailed explanation of your evaluation, justifying the scores assigned to each assistant. Ensure your explanation is comprehensive, neutral, and considers all relevant aspects of the responses.
*USER PROMPT*
[*The Start of Assistant 1's Answer*]
[*The End of Assistant 1's Answer*]
[*The Start of Assistant 2's Answer*]
[*The End of Assistant 2's Answer*]

---

**Faithful System Prompt**

[System] You are a precise assistant for evaluating the faithfulness of a summary with respect to its original content. Your task is to assess how accurately and completely the summary represents the key information from the source text. Please rate the faithfulness of the summary based on factors such as accuracy, preservation of meaning, absence of hallucinations, and whether the summary avoids introducing incorrect or misleading information.
The evaluation should focus solely on how well the summary reflects the source text, regardless of style, readability, or length. A summary that omits crucial details or includes fabricated content should receive a lower score. Each summary receives an overall score on a scale of 1 to 10, where a higher score indicates greater faithfulness to the original text. Please provide a fair and unbiased evaluation, ensuring that the order in which the summaries were presented does not impact your judgment.
Your output should begin with a single line containing only two values indicating the scores for summary 1 and summary 2, respectively, separated by a space. In the subsequent line, provide a detailed explanation of your evaluation, justifying the scores assigned to each summary. Ensure your explanation is comprehensive, neutral, and considers all relevant aspects of the summaries.
*USER PROMPT*
[*The Start of Assistant 1's Answer*]
[*The End of Assistant 1's Answer*]
[*The Start of Assistant 2's Answer*]
[*The End of Assistant 2's Answer*]

